# The Impact of Air Pollution Exposure on the MicroRNA Machinery and Lung Cancer Development

**DOI:** 10.3390/jpm11010060

**Published:** 2021-01-19

**Authors:** Michal Sima, Andrea Rossnerova, Zuzana Simova, Pavel Rossner

**Affiliations:** 1Department of Nanotoxicology and Molecular Epidemiology, Institute of Experimental Medicine CAS, Videnska 1083, 142 20 Prague, Czech Republic; michal.sima@iem.cas.cz (M.S.); zuzana.simova@iem.cas.cz (Z.S.); 2Department of Genetic Toxicology and Epigenetics, Institute of Experimental Medicine CAS, Videnska 1083, 142 20 Prague, Czech Republic; andrea.rossnerova@iem.cas.cz

**Keywords:** air pollution, biomarker, exposure, human, lung cancer, miRNA

## Abstract

Small non-coding RNA molecules (miRNAs) play an important role in the epigenetic regulation of gene expression. As these molecules have been repeatedly implicated in human cancers, they have been suggested as biomarkers of the disease. Additionally, miRNA levels have been shown to be affected by environmental pollutants, including airborne contaminants. In this review, we searched the current literature for miRNAs involved in lung cancer, as well as miRNAs deregulated as a result of exposure to air pollutants. We then performed a synthesis of the data and identified those molecules commonly deregulated under both conditions. We detected a total of 25 miRNAs meeting the criteria, among them, miR-222, miR-21, miR-126-3p, miR-155 and miR-425 being the most prominent. We propose these miRNAs as biomarkers of choice for the identification of human populations exposed to air pollution with a significant risk of developing lung cancer.

## 1. Introduction to miRNA

### 1.1. Basic Information on miRNA

Small non-coding microRNA molecules (miRNAs) were the **most studied RNA** throughout the last decade. The history of their research started in 1993 in Victor R. Ambros’ laboratory, during an investigation of the developmental pathways of the soil nematode *Caenorhabditis elegans*, when lin-4 miRNA, the first miRNA, was described [1]. These single-strand, approximately 22 nucleotide long RNA molecules, which play a crucial role in the epigenetic regulation of gene expression, represent a broad group of nucleic acids described in various species including humans. Regarding miRNAs role in the process of effectivity of translation, these molecules attract the interest of numerous researchers from various fields of biomonitoring research. At the end of 2020, more than 27 years after miRNA discovery, almost 112,000 research articles focused on miRNA molecules can be identified in the PubMed database. This intensive research contributes to revealing new miRNAs every year. Their database, including miRNA sequences, is updated in the miRbase biological catalogue [2]. The latest version released in March 2018 (v22) contains sequences from 48,860 mature miRNAs of various species including 2656 molecules relevant to humans [3,4].

Besides mature miRNAs, many immature miRNA molecules are present in cells during their development, as their maturation is a relatively complex process [5,6]. Their **biogenesis** starts in the nucleus by transcription of the primary miRNA (pri-miRNA), which is an approximately 500-3000 bases long molecule created by transcription of the miRNA gene or intron by RNA polymerase II or III. This process is followed by cleavage of pri-miRNA to an approximately 70 base long precursor miRNA (pre-miRNA) by the Drosha-Pasha (DGCR8) complex. The steps that follow in cytoplasm are started by the export of pre-miRNA from the nucleus by the protein Exportin 5. The next step of maturation; the cleavage of pre-miRNA hairpin to miRNA duplex form, is assisted by RNase III enzyme Dicer, bound to the dsRNA-binding TRBP protein. Finally, this double stranded RNA duplex is transformed into a functional, mature single stranded form of miRNA together with Argonaute proteins. Formation of the RNA-induced silencing complex (RISC) with mature miRNA follows. It has a crucial role in miRNA function related to mRNA degradation in the case of perfect complementarity, or inhibition of translation in the case of non-complementarity of thereof.

miRNA nomenclature has evolved to distinguish mature sequences (denoted miR-XX) from precursors (mir-XX), as well as mature identical sequences originating from different genes (miR-XX-1; miR-XX-2). Adding a lower case letter (a, b…) at the end of the molecule name indicates a close relation among miRNAs with the same number differing only by one or two nucleotides (miR-XXa/b). The -3p or -5p suffix at the miRNA name indicates that miRNAs are excised on the 3′ or 5′ end of the same precursor, respectively (miR-XX-3p; miR-XX-5p) [7].

### 1.2. Methodological Approaches for miRNA Investigation

Along with knowledge of the processes of miRNA maturation and expression, two important aspects should be considered during the planning of human biomonitoring studies. First, the selection of biological material used in the particular study is crucial, as the expression machinery substantially differs between tissue and biological fluids, as plasma or urine. Second, a selected methodological approach can impact the overall interpretation of the results. To date, three major **methodological strategies** of miRNA analysis have been used. They are based on amplification, hybridization, or sequencing protocols [8,9,10]. Their choice for individual studies strongly depends on the aim of the experiment, the quality and quantity of samples as well as on the budget of the researcher.

Among **amplification-based** approaches, quantitative real-time polymerase chain reaction (qRT-PCR) is the most available method, which is still considered the gold standard due to its specificity, accuracy, sensitivity, and relatively low price. This approach can be used for individual miRNA detection, as well as for predefined sets of a few hundred molecules in an array format. Two variants of qRT-PCR differing in cost, nucleotide labeling and specificity are commonly used: CYBR Green or TaqMan. The CYBR Green approach is cheaper but there is a possibility of non-specific dsDNA-fluorescent dye binding which may negatively affect the results. In contrast, the TaqMan assays work based on a dual labeled oligonucleotide and exonuclease activity of Taq polymerase enzyme which increase the specificity [11]. However, to assure accuracy, for both qRT-PCR variants normalization of the data to the expression of an internal reference gene is mandatory. The reference gene, usually a housekeeping or another constitutively expressed gene, should be stably expressed in different cell types and under various experimental and treatment conditions [12].

The **hybridization method** represents a more advanced approach that is based on binding of miRNAs in a sample to specific complementary probes immobilized on surface of glass slides (microarrays). The current microarrays available on the market allow for the detection of a relatively high number of human miRNAs included in the previous version of miRNA release (v21 = 2549 miRNAs). However, this approach is limited to the already described miRNAs only. Additionally, due to the risk of false-positive results, verification of the data by qRT-PCR is required. The **sequencing strategy** by next generation sequencing (NGS) allows for the analysis of a full set of small RNA including miRNAs presented in a sample with the possibility to discover novel miRNAs, or other non-coding RNAs. The relatively high cost and demanding data processing could be a limitation for application in some studies.

### 1.3. miRNA as a Biomarker

Specific miRNA pattern has been repeatedly used as a biomarker of various diseases, including cardiovascular, neurodegenerative, or retinal disorders, as well as cancer [13,14,15]. The presence of specific miRNAs serves as an important diagnostic, prognostic, and therapeutic marker especially in relation to various **cancers**. Even though this record only started in 2002 when the deregulation of miR-15 and miR-16 was described in patients with chronic lymphocytic leukemia (CLL) [16], more than 53,000 studies have already been published according to a PubMed database search for keywords: “miRNA” and “cancer”.

Various environmental and chemical **exposures** affect humans on a daily basis. These stressors are considered important risk factors for disease development. According to the World Health Organization (WHO), air pollution exposure is considered the greatest environmental risk factor of ill health [17]. The most recent data (related to year 2016) estimates that 4.2 million premature deaths occur each year due to outdoor air pollution and 3.8 million deaths are related to household air pollution. Among them, the majority were associated with ischemic heart disease and stroke, pneumonia, chronic obstructive pulmonary disease, and lung cancer which are estimated to account for more than half a million cases per year. These facts along with new methodology development, resulted in investigation of the epigenetic markers, including DNA methylation, miRNA expression or histone modification.

Similar to the research on diseases, many exposure studies linked to various environmental stressors revealed a specific miRNA expression pattern. The number of these studies has also increased during recent years. More than 3000 reports can be found using PubMed database search for keywords “miRNA” and “exposure”.

Nowadays, links between **environmental exposure and risk of cancer related to deregulation of specific miRNAs** have also been described. Even though a huge number of publications related to miRNA, environmental exposure and cancer have been published and reviewed [18,19,20,21,22,23], we attempted to go deeper into these topics and concentrated in detail on the narrower part of this research. The main **aim of this review** was to find an intersection of specific miRNAs expressed in relation to (i) lung cancer, as the most common cancer related to air pollution exposure (see Section 2), (ii) air pollution exposure which is relevant to human populations living in polluted areas (see Section 3) and identify the specific miRNA expression changes related to air pollution and potentially leading to lung cancer. To fulfil the aim, we focused on particular exposure conditions in studied populations (chronic, acute, or seasonally changed) including the concentrations of environmental stressors (e.g., particulate matter (PM) of various aerodynamic diameter), the age of the studied human population, as well as the methodological approach used for miRNA pattern investigation due to their different complexity.

## 2. Lung Cancer and miRNA

Since the discovery of the relationship between miRNA deregulation and CLL [16], many research groups have focused their attention to investigation of the connection between miRNA and various cancers (reviewed by [24]).

The mechanisms of miRNA deregulation in cancer are frequently linked with alterations in genomic miRNA copy number and gene locations. There are also other processes, which could influence miRNA expression, such as dysregulation of key transcription factors, epigenetic modulation, or mutation or aberrant expression of any component of the miRNA biogenesis pathway (reviewed by [25]). In cancer, miRNAs can serve as oncogenes or tumor suppressors. The miRNA-suppressors inhibit oncogenes and/or apoptosis- or cell differentiation-controlling genes which results in tumor suppression. On the contrary, the miRNA-oncogenes support tumor development usually by inhibiting tumor suppressor genes and/or genes involved in cell differentiation or apoptosis (reviewed by [26]). The role of miRNAs (e.g., miR-126, miR-221, miR-222) in angiogenesis, an important process associated with progression of several diseases, including cancer has also been reported. During angiogenesis, endothelial cells are activated and proliferate resulting in formation of tubular structures and supporting tumor growth [27].

In 2004, the role of miRNA in human lung cancer was highlighted and connected to shortened postoperative survival [28]. WHO classified lung cancer into two clinicopathological categories–contentious and intermittently metastatic small cell lung cancer (SCLC) and the more prevalent but less destructive non-small cell lung cancer (NSCLC) [29]. Even if new detection and therapeutic methods are in progress, lung cancer is often diagnosed in the later stage with survival rate around 20% [30].

In 2015, Feng et al. reviewed the role and importance of miRNA deregulation for lung cancer diagnostics and possible treatment. From previously published papers, they concluded, that several miRNAs are highly expressed in NSCLC patients when compared to healthy individuals but, on the other hand, other miRNAs are specifically under-expressed in lung cancer cases [31]. Based on that, miRNAs could serve as biomarkers for early detection of NSCLC which in effect could decrease the high risks of lung cancer deaths.

The searching of PubMed database for keywords “miRNA AND lung cancer AND human cases” produced 284 hits. In order not to repeat the previously summarized data, from the review mentioned above [31] until the present time, the number of publications was reduced to 158. From this amount, we further excluded review articles, meta-analyses, non-human or cell-based-only studies or studies where only cancer patients (without controls) were involved. After this specification, 54 papers met the conditions: miRNA expression was compared between lung cancer patients and healthy subjects (or other than tumor tissue from NSCLC patients).

Overall, four various input materials were used for miRNA detection: blood (4 studies), plasma (6 studies), serum (10 studies), and, the most common, tumor tissue (34 studies). Patniak et al. (2017) suggested, that whole blood is probably not suitable for the later miRNA quantification due to a lack of differences in expression levels based on microarray and qRT-PCR [32]. In disagreement with this finding, two studies observed deregulated miRNA in LC patients [33,34]. In the majority of reports (47) researchers utilized qRT-PCR for the quantification of miRNA levels, in the remaining studies, microarrays [35,36,37] or the novel method of PCR-droplet digital PCR [38,39,40] were used. For one study, only the abstract, without this information, was available [41].

In addition to being a biomarker, which could reveal the early stage of LC, in more than twenty publications the role of miRNAs in lung cancer development has been proposed. Some miRNAs have been described as **oncogenes**: miR-675 was associated with NSCLC progression through activation of nuclear factor-κB signaling pathway [42], miR-198-5p was downregulated in the early stage of lung squamous cell carcinoma and could play an important role via its target genes [43]. The metastasis suppressor 1 was repressed by miR-29a which resulted in tumor proliferation [44], leucine zipper putative tumor suppressor 3 was deregulated due to its connection to miR-1275 [45] and the level of miR-99a-5p was connected to poor survival in surgically resected lung adenocarcinoma specimen [46]. In the pulmonary adenocarcinoma, miR-210 and miR-183 were upregulated and served as oncogenes [47].

Alternatively, the **onco-suppressor** role has been described for miR-218-5p, miR-497, miR-34c due to their inhibition of cancer cell proliferation and migration [47,48,49], as well as for miR-451 and its link with macrophage migration inhibitory factor [50]. The onco-suppression was further linked with miR-219 that targets the high mobility group AT-hook 2 [51], and with miR-504 that is upregulated and inhibits cell invasion and proliferation [52]. The signal transducer and activator of transcription-3 is the direct target of miR-454 [53] and the transforming growth factor β receptor 2 is downregulated by miR-107 [54], other onco-suppressors.

Twelve miRNAs have been specifically proposed as **possible therapeutic targets**: miR-34b-3p that targets cyclin-dependent kinase 4 [37]; miR-588 whose silencing causes the increased expression of prostaglandin [55]; miR-103 that deregulates the programmed cell death 10 [56]; miR-491-5p that might reduce the expression of matrix metallopeptidase 9 [57]; miR-140-5p whose restoration may support the current LC therapies [58]; miR-12528 that controls the insulin-like growth factor 1 receptor, which is overexpressed in most of the cancer types [59]; miR-1260b that acts as onco-miRNA when inhibiting protein tyrosine phosphatase receptor type kappa and therefore might serve as a novel target for treatment [60]. The connection of p53 tumor-suppressor and miR-101 is important for tumor suppression due to the link with nucleolar stress [61]. miR-196b-5p is involved in Quaking-GATA binding protein 6-tetraspanin 12 pathway [62]. SRY-Box transcription factor 18 and its mRNA levels are influenced by the deregulation of miR-7a and miR-24-3p [38,39] and the expression of RUNX family transcription factor 2 is connected to miR23-b [63]. Therefore, their deregulation is suggested as a potential therapeutic strategy.

Altogether, in the 28 remaining studies, 97 various miRNAs were suggested as being biomarkers, which could help to reveal lung cancer in the early stages leading to a possible survival rate increase. In the following paragraph, several studies with the most commonly detected differentially expressed miRNAs are described. The total overview is summarized in Table 1.

Wozniak et al. screened 754 miRNAs in 100 LC patients and a corresponding control group and developed a 24-plasma miRNA panel, which was capable of distinguishing these study groups based on differential miRNA expression [35]. A similar study was performed by Wang et al., where levels of five miRNAs were elevated after comparison of patients and healthy individuals [75]. Twenty miRNAs from plasma could be used as the diagnostic classifier for lung adenocarcinoma [36] and the combination of four miRNAs was validated out of 21 molecules as the microRNA expression signature for the LC patients [72]. Niu et al. (2018) detected ten differently expressed miRNAs when LC patients and healthy subjects were compared. Based on their results, some of these miRNAs were associated with adenocarcinoma or squamous cell carcinoma [67]. With more than 90% specificity and sensitivity, four plasma miRNAs combined together could serve as a reliable tool for LC diagnostics even in the early stage of the disease [65]. By microarray, 338 differently expressed miRNAs were detected in blood from LC patients and later, after evaluating in larger sample groups and using qRT-PCR, four of them were chosen as promising diagnostic instruments [33]. The most recent study focused on searching for the appropriate biomarkers for lung tumor detection was published in 2020. Thirty-five miRNAs were indicated as biomarkers with different expression in LC patients, and, after validation in three additional cohorts, a five miRNA panel was created [64].

As described in this review and shown previously [31], the miRNA-lung cancer link is well established. Studying this relationship has revealed, that production of miRNA is influenced by the lung tumor and the progression of the lung tumor is dependent on the miRNA levels as well. Some miRNAs serve as oncogenes, others as tumor suppressors. Some miRNA levels are upregulated and some downregulated, which influences the protein translation and tumor progression/suppression. In conclusion, several miRNAs could be used as an early diagnostic tool which might improve the lung cancer prognosis and because of their connection to protein production, some of them have been proposed as being a therapy target for individuals, who suffer from this disease.

## 3. Air Pollution and miRNA

In comparison with lung cancer, the link between miRNA expression and air pollution exposure has been less studied. The first reports focusing on this investigation were published in 2012. In this section we aim to summarize the current knowledge on deregulation of miRNA expression in human subjects exposed to various types of air pollutants. In contrast to the miRNA-lung cancer link discussed above, we present the topic in more detail, as the review literature on this topic is lacking. Although the latest review article on miRNAs as biomarkers of exposure to environmental pollutants was published in 2019, the authors focused specifically on the role of the environment without investigating the miRNA-air pollution-lung cancer relationship [86].

To identify studies that have focused on the investigation of air pollutants on the modulation of miRNA expression in humans, we searched the PubMed database for the string “miRNA air pollution” and limited the output to “Humans” as a species. This query yielded 97 results which were further checked to obtain the reports that analyzed miRNA expression in human subjects exposed to any type of air pollutant. Only studies that involved healthy subjects, or alternatively diseased participants not suffering from cancer were further considered. We also excluded review articles from our search. As a result, we identified 27 studies published between 2012 and 2020 focused on miRNA expression in humans exposed to various air pollutants (Table 2).

A total of 18 reports focused on the investigation of the effects of particulate matter (PM) of various aerodynamic diameter (PM10, PM2.5, ultrafine particles (UFP)), often along with other traffic- or combustion-related pollutants (NOx, CO, CO_2_, black carbon (BC)). In 5 studies, the effects of tobacco/cigarette smoke were investigated, while other pollutants (e.g., liquid petroleum gas (LPG) and diesel exhaust, wood smoke, volatile organic compounds, ozone) were evaluated in 4 publications. The analytical methods included various variants of qRT-PCR (17 studies), microarrays and other hybridization-based approaches (8 studies) and NGS (2 reports). Most of the studies (21) investigated miRNA expression in blood-derived material (whole blood, serum, plasma, extracellular vesicles), other matrices included placenta (1×), saliva (1×), lung tissue (1×), spermatozoa (1×), bronchoalveolar lavage (1×) and sputum (1×). The data were obtained for 4940 subjects, that mostly included healthy participants of various age, but some suffered from heart disease, chronic obstructive pulmonary disease (COPD), or were atopic.

### 3.1. The Effect of Air Pollution on miRNA Expression in Healthy Adults

The majority of studies reported miRNA expression changes after exposure to air pollutants in general adult populations. Thus, the effects of PM2.5, UFP, black carbon and soot on miRNA expression were investigated in a multi-centric study among 143 healthy volunteers living in Switzerland, United Kingdom, Italy, and the Netherlands. The authors used the microarray technology to identify a total of seven microRNAs (miR-24-3p, miR-4454, miR-4763-3p, miR-425-5p, let-7d-5p, miR-502-5p, miR-505-3p) extracted from whole blood to be correlated with exposure to PM2.5. Interestingly, the effect of other pollutants was not significant [87]. Another study that involved 24 healthy subjects exposed to air pollutants during physical activity and the resting phase used NGS technology to correlate miRNA present in blood plasma with exposure to PM10, PM2.5, NO, NO_2_, CO, CO_2_, BC and UFP. Although the exposure to a mixture of the pollutants affected expression of nine miRNAs (miR-28-3p, miR-222-3p, miR-146-5p, miR-30b-5p/30c-5p and miR-320a-3p/320b/320c/320d/320e were positively associated; miR-532-5p, miR-192-5p/215-5p, miR-144-3p and miR-425-5p showed a negative relationship), no specific effects of PM2.5 and PM10 were detected. However, the effects of NO, NO_2_, CO, CO_2_, BC and UFP were observed [89]. In another report, a total of 24 non-smoking subjects (healthy, or suffering from ischemic heart disease (IHD), or COPD) were exposed to various levels of ambient air pollution and miRNA expression in blood plasma was assessed using NGS. The authors identified 54 circulating miRNAs associated with exposure to PM10, PM2.5, black carbon, UFP and NO_2_ following only 2h exposure to air pollution. These molecules have been described as being related to negative consequences of traffic pollutants in the lung, heart, kidney and brain [92]. The effect of short (2h) PM10, PM2.5, NO, NO_2_, CO, CO_2_, BC and UFP exposure on miRNA expression in whole blood was further investigated using microarray technology among a total of 89 volunteers, including healthy subjects and those with COPD and IHD. The investigated populations originated from two cohorts with different levels of air pollution. The authors found miR-197-3p, miR-29a-3p, miR-15a-5p, miR-16-5p and miR-92a-3p linked with the exposure scenarios, although the expression of individual molecules was cohort-specific with little overlap between both sets of samples. These miRNAs play a role in cancers and Alzheimer’s disease indicating a health risk associated with exposure to air pollutants. An effect of COPD and IHD on miRNA expression profiles was not found [93]. The potential role of PM2.5 on cytokines associated with systemic inflammation was assessed in 55 healthy volunteers exposed to different levels of the pollutant. A negative correlation of the exposure with miR-21-5p, miR-187-3p, miR-146a-5p, miR-1-3p and miR-199a-5p expression in whole blood confirmed the role of cytokines in response to exposure to air pollution [94]. Another study in 22 healthy subjects focused on the long-term effects of ambient PM2.5 exposure on miRNA expression in extracellular vesicles in serum involved in pathways related to cardiovascular diseases. The authors detected increased levels of miR-126-3p, miR-19b-3p, miR-93-5p, miR-223-3p, miR-142-3p, miR-23a-3p, miR-150-5p, miR-15a-5p and miR-191-5p let-7a-5p that are linked to oxidative stress, inflammation and atherosclerosis [97]. Targeted analysis of miR-21, miR-222 and miR-146a in the blood of 50 healthy subjects exposed to environmental levels of PM10 was performed using qRT-PCR in another study. An increase in PM10 concentrations was associated with a decrease of miR-21 and miR-222 expressions that are involved in inflammatory and oxidative stress pathways [100]. The specific role of metals in PM2.5 was studied in 120 healthy subjects exposed to moderate air pollution by measuring the expression of miR-4516 in serum. The expression of this miRNA was positively associated with Al, Pb and Cu levels suggesting an important role of miR-4516-autophagy pathway in response to PM2.5 and PM-associated metals. In a study involving 60 truck drivers and 60 office workers living in the highly polluted city of Beijing, the effect of PM2.5, PM10 and elemental carbon (EC) on miRNA expression in whole blood was analyzed using a hybridization technology. Interestingly, no consistent significant effects of either PM2.5, or PM10 exposure was observed. PM10 affected the expression of 12 miRNAs in office workers only, while short-term EC exposure had significant impacts on 28 miRNAs in office workers and 29 miRNAs in truck drivers, although only 5 miRNAs were common in both groups (miR-125a-5p, miR-1274a, miR-600, miR-1283, miR-10a). The deregulated miRNAs seem to play a role in the immune response [102].

### 3.2. The Effect of Air Pollution on miRNA Expression in Children

The effect of prenatal exposure to air pollutants on miRNA expression in placenta tissue was investigated by qRT-PCR by Tsamou et al. [90]. In a group of 210 newborns placenta tissue was collected upon delivery and mothers’ exposure to PM2.5 in individual trimesters was correlated with expression levels of miRNA. The results showed an inverse relationship of PM2.5 levels in the 2nd trimester with miR-21, miR-146a and miR-222 expression, while miR-20a and miR-21 levels were positively associated with air pollution in the 1st trimester. The common putative target of these miRNAs is PTEN (tumor suppressor phosphatase and tensin homolog) that is involved in the pathways regulating cell survival, cell cycle, angiogenesis and metabolism suggesting the impact of PM2.5 exposure on these processes.

Research on environmental exposure to PM and other air pollutants in children was conducted by Liu et al. [88] and Vriens et al. [96]. The first study focused on a link between air pollution and childhood asthma. In a group of 180 asthmatic and 180 healthy children, the serum levels of miR-155 were analyzed by qRT-PCR and their correlation with HCHO, NO_2_ and PM10, PM2.5 and PM1 was assessed. In asthmatic children the levels of miR-155 were significantly higher and were associated with indoor PM2.5 and HCHO concentrations. As this miRNA plays an important role in asthma progression, indoor air pollution seems to be involved in aggravation of the disease in this study group [88]. In a group of 80 healthy children, saliva was collected, expression of miR-222 and miR-146a assessed by qRT-PCR and link with recent exposure to PM2.2 and UFP investigated. While a positive correlation with UFP concentrations was detected for miR-222 levels, which was reported to participate in cell cycle regulation, no such effects were found for miR-146a [96].

### 3.3. The Effect of Air Pollution on miRNA Expression in Elderly Subjects

Two studies focused on miRNA expression associated with air pollution in elderly men originating from the Normative Aging Study [91,104]. In a small group of 22 subjects, exposure to PM2.5 was linked with increased blood pressure and positively associated with miR-199a/b and miR-223-3p expression in extracellular vesicles. The expression of miR-199a/b was further affected by DNA methylation near the enhancer region of the gene encoding this molecule. Both miRNAs seem to target proteins implicated in important cardiovascular functions [91]. The potential effect of PM2.5, black carbon, organic carbon, and sulphates on the expression of fourteen candidate miRNAs in blood leukocytes was investigated among 153 subjects. A negative correlation between pollutant levels and miR-1, miR-126, miR-135a, miR-146a, miR-155, miR-21, miR-222 and miR-9 was detected. The strongest link was found for 7-day moving averages of PM2.5 and black carbon, and 48-h moving averages for organic carbon. The deregulated miRNAs most likely participate in HMGB1/RAGE signaling pathway that is associated with the enhanced expression of proinflammatory cytokines [104].

### 3.4. The Effect of Air Pollution on miRNA Expression in Overweight/Obese Subjects

The role of miRNA expression in the risk of cardiovascular disease modified by exposure to PM10 in overweight/obese subjects was reported in two studies [95,99]. A larger investigation of 1630 subjects showed downregulation of let-7c-5p, miR-106a-5p, miR-143-3p, miR-185-5p, miR-218-5p, miR-331-3p, miR-642-5p, miR-652-3p and miR-99b-5p expression in extracellular vesicles after short-term exposure to PM10. These miRNAs exhibit a putative role in cardiovascular disease and mediate changes of fibrinogen levels associated with PM10 exposure suggesting a role of PM in increased coagulation [95]. In another study, decreased miRNA expression in the peripheral blood of 90 obese subjects was found after exposure to PM10 48 h before sample collection. These miRNAs included miR-145, miR-197, miR-30b, miR-345, miR-26a, miR-425-5p, miR-331, miR-140-3p and miR-101. PM10 exposure was associated with a blood pressure increase further modulated by miRNA-101 expression [99]. These reports indicate that miRNA expression represents a molecular mechanism underlying the effects of air pollution on blood pressure.

### 3.5. The Effect of Occupational Exposure to Polluted Air on miRNA Expression

miRNA expression was also assessed in occupationally exposed subjects. In extracellular vesicles of healthy steel plant workers miRNA levels were measured by qRT-PCR [98,103]. Among 55 subjects, 17 miRNAs were found to be affected, including mir-196b, miR-302b, miR-200c, miR-30d. The pathway analysis revealed the role of mir-196b in insulin biosynthesis; miR-302b, miR-200c, miR-30d were related to inflammatory and coagulation markers. Thus, inhalation exposure to PM with metallic components may have adverse cardiovascular and metabolic effects [98]. In a study of 63 workers, increased expression of miR-128 and miR-302c after 3 days exposure to PM was detected. Pathway analysis identified miR-128 as a part of coronary artery disease pathways, and miR-302c to be involved in coronary artery disease, cardiac hypertrophy and heart failure pathways [103]. Both studies thus highlight the role of PM exposure in negative impacts on the cardiovascular system.

### 3.6. The Effect of Tobacco Smoking on miRNA Expression

The effect of cigarette/tobacco smoke exposure on miRNA expression was studied among healthy subjects, those suffering from COPD, as well as in mothers and newborns. While COPD is a pulmonary disease linked with genetic and environmental factors, dysregulation of miRNAs has also been shown to play a role. The effect of tobacco smoke exposure in miRNA deregulation was investigated in serum and lung tissue of a total of 84 subjects [105,107]. Serum levels of miR-22-3p were upregulated in smoking COPD subjects when compared with COPD subjects exposed to biomass smoke. Non-exposed or healthy controls were not included in this study. miRNA-22-3p was suggested to activate antigen-presenting cells in lungs in relation to tobacco smoke exposure [105]. Another study revealed downregulation of miR-181c in lung tissues from smoking patients with COPD when compared with subjects who had never smoked. Overexpression of this miRNA decreases inflammatory response, neutrophil inflammation, ROS generation and inflammatory cytokines production [107]. In 775 healthy subjects, cigarette smoking was associated with expression of miR-29a, miR-93, let-7a, and let-7g using a machine learning approach suggesting these molecules as potential serum biomarkers of environmental tobacco smoke exposure [106]. In spermatozoa of 7 non-smokers and 6 smokers miRNA profiling revealed differences in the expression of 28 miRNAs that were shown to be involved in several pathways, including cellular proliferation, differentiation and death, as well as reproductive system disease [109]. The effect of prenatal cigarette smoke exposure on the expression of miR-155 and miR-223 was studied in the maternal and cord blood of 441 mothers/newborns pairs. A positive correlation was found between miR-223 expression and maternal urine cotinine levels, indoor concentrations of benzene and toluene. The effects were not observed for miR-155. The results indicate a role of miR-223 expression on regulatory T cell numbers in the cord blood with a subsequent allergy risk to children of mothers exposed to tobacco smoke [108].

### 3.7. miRNA Expression in Subjects Exposed to Other Sources of Air Pollution

The next paragraph summarizes the results from populations that do not fit to the above-reported groups. The effects of household air pollution on miRNAs associated with inflammatory response was studied among 52 healthy women who used wood and LPG for cooking. Specifically, the expression of miR-126 and miR-155 was assessed in plasma by qRT-PCR and the results correlated with 1-hydroxypyrene levels, as a marker of smoke exposure. The expression of both molecules was significantly higher in the subjects exposed to wood smoke. As the analyzed miRNAs are important modulators involved in vascular dysfunction and atherosclerosis, the results indicate a greater health risk associated with burning wood than using LPG [110]. In another study, fifteen atopic subjects were exposed for 2 h to filtered air or diesel exhaust followed by bronchial allergen challenge in a controlled study and the expression of miRNA was assessed in bronchoalveolar lavage. Diesel exhaust induced expression of a greater number of miRNAs when compared with the controls. The presence of allergen significantly modulated the expression of miR-183-5p, miR-324-5p and miR-132-3p, while diesel exhaust alone did not have this effect. Negative correlations were observed between miR-132-3p and CDKN1A, a regulator of cell cycle progression in G1, as well as miR-183-5p and HLA-A, human leukocyte antigens [111]. The impacts of exposure to volatile organic compounds (VOC), including toluene, xylene and ethylbenzene was investigated in 50 healthy occupationally exposed subjects and controls. miRNA expression was assessed in whole blood using microarray technique. Specific signature of exposure to individual compounds was found: expression of 467 miRNAs was associated with toluene exposure, 211 miRNAs with xylene exposure and 695 miRNAs with xylene inhalation. These signatures may serve as indicators of VOC exposure. However, identification of the potential mechanisms underlying the exposure was not performed in this study. The impacts of inhalation exposure to ozone on miRNA expression in human bronchial airways were investigated by Fry et al. [113]. Twenty healthy subjects were enrolled and exposed for 2 h. Sputum samples were collected 48 h pre-exposure and 6 h post-exposure and miRNA expression was assessed by microarrays. Ozone exposure increased the expression of 10 miRNAs (miR-132, miR-143, miR-145, miR-199a, miR-199b-5p, miR-222, miR-223, miR-25, miR-424, and miR-582-5p). Pathway analysis revealed, among other biological functions and properties, their link with inflammation and immune-related diseases.

### 3.8. miRNAs Commonly Deregulated by Various Types of Air Pollutants

Due to the diversity of the biological material used for the detection of miRNA expression, various analytical methods, characteristics of human subjects and exposure conditions, identification of commonly deregulated miRNA(s) that may serve as biomarker(s) of exposure to air pollutants is rather difficult. From the studies discussed in this review, miR-222 was found to be affected by air pollutant exposure in six articles, followed by miR-223 family (mir-223 and miR-223-3p), miR-21 family (miR-21 and miR-21-5p) and miR-155, each reported in four studies, and miR-126 family (miR-126 and miR-126--3p) and mir-425-5p, each described in three publications. Other miRNAs commonly appeared in two studies only or were unique for a single report.

## 4. miRNA Affected by Air Pollution Exposure and Implicated in Lung Cancer

In this section, we discuss the miRNAs that we identified to be commonly associated with air pollution exposure and lung cancer risk. Such miRNAs deserve the most attention as promising biomarkers that may inform of exposure to harmful pollutants potentially contributing to lung cancer development. An overview of the commonly deregulated miRNAs is provided in Table 3 and Figure 1; the most prominent molecules are discussed further in this section.

**miR-222** appears to be deregulated in many cancers [115], including NSCLC [114] in which it targets tumor suppressors (PTEN, TIMP3) and enhances cellular migration by the activation of the AKT pathway. The molecule is involved in several steps of carcinogenesis, as e.g., tumor cell invasion and metastasis, regulation of apoptosis and drug resistance and the induction of tumor angiogenesis. Our literature review indicates that its expression was also associated with exposure to mixtures of air pollutants, specifically to PM10, PM2.5, UFP, black carbon, organic carbon, sulphates, and ozone. This link was described in six studies making miR-222 the molecule that is most commonly deregulated in the context of air pollution exposure.

**let-7a-5p** is involved in lung cancer development, most likely by targeting *BCL2L1*, *IGF1R*, *MAPK8*, and *FAS* genes thus affecting cell proliferation, growth arrest and apoptosis, as well as production and elimination of reactive oxygen species [116]. Its expression was found to be affected in subjects exposed to cigarette smoke and PM2.5. A variety of activities of **miR-21** have been described, including functions as an oncogene, that inhibits apoptosis, promotes cell differentiation and interstitial fibrosis. Its role in hypertension and lung cancer has also been reported [117,118]. In addition, its expression is affected by exposure to air pollutants (PM2.5 and PM10, black carbon, organic carbon, and sulphates) as reported in four studies. **miR-29a** was detected to be overexpressed in NSCLC tissues, where it was positively associated with malignancy of the disease and negatively associated with survival. The miRNA targets the *MTSS1* gene that encodes the protein inhibiting cell migration and proliferation [44]. Exposure to PM of various sizes, as well as to NOx, CO, CO_2_, and cigarette smoke were factors involved in the deregulation of miR-29a expression in environmentally exposed subjects. **miR-93** plays a role in several human cancers, including lung cancer, although its effect can be inconsistent, as it may act both as an oncogene and tumor-suppressor. The target genes of this miRNA were reported to be closely related to transcription with *MAPK1*, *RBBP7* and *Smad7* being the hub genes [119]. The expression of miR-93-5p was affected in human subjects exposed to PM2.5 and cigarette smoke. Circulating **miR-126-3p** was associated with exposure to asbestos and with malignant mesothelioma [120], although in another study it suppressed the progression of NSCLC [121]. Low levels of miR-126-3p were associated with poor pathological stage, large tumor diameter and lymph node metastasis in lung adenocarcinoma. This miRNA was suggested to regulate the pathways involved in apoptosis and cancer [122]. PM2.5, black carbon or wood smoke were prominent pollutants affecting levels of this molecule. Expression of both strands of **miR-145** is downregulated in lung cancer where these molecules regulate cell cycle pathway genes and significantly reduce patient survival [123]. In human exposure studies, PM10 and ozone exposure were associated with expression of this molecule. **miR-155** is involved in a variety of processes linked to immunity, inflammation, and hematopoiesis. Its aberrant expression was observed in malignant and non-malignant diseases affecting the nervous, immune and cardiovascular system [124]. This molecule is also deregulated in lung disorders, including asthma and cystic fibrosis and lung cancer [125]. Similar to miR-126-3p, its expression was modified by PM2.5, black and organic carbon, as well as wood smoke exposure. These pollutants further affected the expression of **miR-425-5p** that appears to act as an oncogene in lung cancers, including squamous cell carcinoma [126] and NSCLC [127] in which the overexpression of the molecule was associated with poor prognosis.

In contrast to previously mentioned miRNAs, **miR-223-3p** may have a function as a tumor-suppressor. The molecule is involved in inflammatory processes, it targets inflammasome components affecting the development of autoimmune diseases [128]. It also regulates the expression of GLUT4, a protein whose expression is altered in prediabetes and diabetes [129]. miR223-3p was further detected to be overexpressed in neutrophils of patients with asthma [130]. A recent study reported tumor-suppressing effects of this miRNA in lung squamous cell carcinoma [131]. Considering these results, deregulation of miR-223-3p following PM2.5 and ozone inhalation reflects the role of air pollutants in the development of immune system-related disorders rather than cancer.

## 5. Conclusions

In this review we aimed to summarize the current state of knowledge on miRNAs deregulated in lung cancer and miRNAs affected by exposure to air pollutants. As exposure to air pollution represents a dominant factor in the development of lung cancer and other respiratory system disorders, we further intended to identify the miRNAs commonly affected by both conditions. Such molecules could serve as biomarkers of choice for identification of human populations in greater risk of lung cancer resulting from exposure to air pollution. Our literature search identified a total of 25 miRNAs that meet such criteria. Among them, miR-222, miR-21, miR-126-3p, miR-155 and miR-425 may be considered the prominent molecules, as they were identified to be deregulated in multiple studies. PM2.5 is an air pollutant commonly affecting the expression of molecules. Thus, our observation is in agreement with the classification of air pollution as a human carcinogen [132]. It should however be noted that the number of studies investigating the link between air pollution and miRNA expression is limited when compared with the studies on cancer-miRNA relationship, and the methods used for detection of miRNA expression widely differ. Additionally, the effect of various confounders, including e.g., age of human subjects, lenght of exposure, genetic variability associated single nucleotide polymorphisms (SNPs) in genes encoding miRNAs, or the role of epigenetic adaptation should be considered. In particular, the process of epigenetic adaptation, previously reported by us and other authors (reviewed e.g., in [133,134,135]), significantly modifies the environment-organism interactions potentially resulting in a reduction of negative impacts of pollutants on the organism. These facts should be taken into account as they may potentially bias our conclusions.

## Figures and Tables

**Figure 1 jpm-11-00060-f001:**
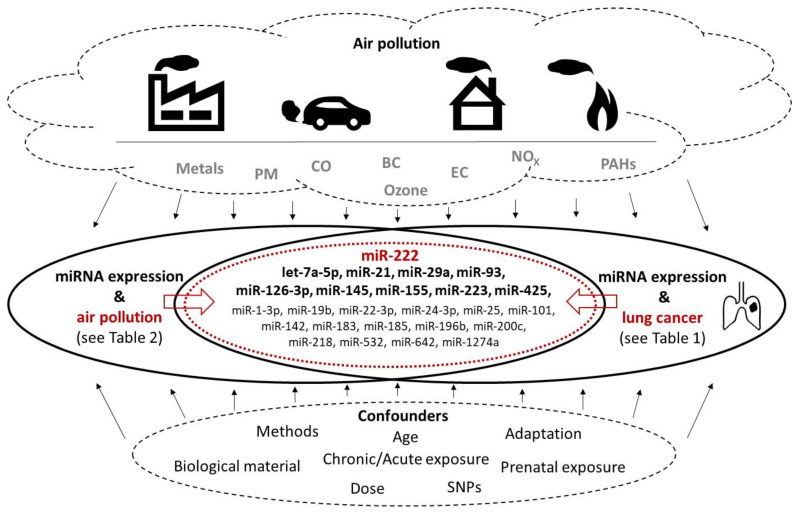
Graphical presentation of miRNAs commonly deregulated in lung cancer and human subjects exposed to various air pollutants. Factors affecting the results (types of air pollutants, methodological and other confounders) are also shown. See Table 3 for more details.

**Table 1 jpm-11-00060-t001:** An overview of studies focused on relationships between lung cancer and miRNA expression.

miRNA	Tissue	Patients	Main Output	Method	Reference
let-7a-5p, miR-214-3p, miR-1291, miR-1-3p, miR-375	Serum	744	DB	qRT-PCR	[64]
miR-29a-5p, miR-4491, miR-542-3p, miR-135a-5p	Blood	145	DB	Microarray + qRT-PCR	[33]
miR-2114, miR-2115, miR-449c	Blood	NS	DB	qRT-PCR	[34]
miR-210-3p, miR-126-3p, miR-145, miR-205-5p	Plasma	471	DB	qRT-PCR	[65]
miR-25	Plasma	114	DB	qRT-PCR	[66]
miR-26a-5p, miR-126-5p, miR-139-5p, miR-152-3p, miR-200c-3p, miR-3135b, miR-151a-3p, miR-151a-5p, miR-151b, miR-550a-3p	Plasma	437	DB	qRT-PCR	[67]
miR-339-5p, miR-21	Plasma	28	DB	Microarray + qRT-PCR	[68]
miR-532, miR-628-3p, miR-425	Plasma	201	DB	qRT-PCR	[69]
let-7c, miR-122, miR-182, miR-193a-5p, miR-200c, miR-203, miR-218, miR-155, let-7b, miR-411, miR-450b-5p, miR-485-3p, miR-519a, miR-642, miR-517b, miR-520f, miR-206, miR-566, miR-661, miR-340, miR-1243, miR-720, miR-543, miR-1267	Plasma	100	DB	Microarray	[35]
miR-107	Serum	NS	OS	qRT-PCR	[54]
miR-223	Serum	75	DB	ddPCR	[40]
miR-21-5p, miR-140-5p, miR-126-3p	Serum	23	DB	qRT-PCR	[70]
miR-661	Serum	150	DB	qRT-PCR	[71]
miR-23b, miR-423-3p, miR-148b, miR-221	Serum	50	DB	qRT-PCR	[72]
miR-21	Serum	50	DB	NS	[41]
miR-22, miR-126	Serum	127	DB	qRT-PCR	[73]
miR-451, miR-1290, miR-636, miR-30c, miR-22-3p, miR-19b, miR-486-5p, miR-20b, miR-93, miR-34b, miR-185, miR-126-5p, miR-93-3p, miR-1274a, miR-142-5p, miR-628-5p, miR-486-3p, miR-425, miR-645, miR-24	Serum	253	DB	Microarray	[36]
miR-21	Serum	50	DB	qRT-PCR	[74]
miR-483-5p, miR-193a-3p, miR-25, miR-214, miR-7	Serum	221	DB	Microarray + RT-PCR	[75]
miR-196b-5p	Tissue	713	PTT	qRT-PCR	[62]
miR-497	Tissue	15	OS	qRT-PCR	[49]
miR- 661-3p	Tissue	12	DB	qRT-PCR	[76]
miR-99a-5p	Tissue	50	OG	qRT-PCR	[46]
miR-29a	Tissue	55	OG	qRT-PCR	[44]
miR-101	Tissue	200	PTT	qRT-PCR	[61]
miR-21	Tissue	89	DB	qRT-PCR	[77]
miR-1275	Tissue	70	OG	qRT-PCR	[45]
miR-12528	Tissue	20	PTT	qRT-PCR	[59]
miR-182-5p	Tissue	23	DB	qRT-PCR	[78]
miR-491-5p	Tissue	100	PTT	qRT-PCR	[57]
miR-1260b	Tissue	26	PTT	qRT-PCR	[60]
miR-198-5p	Tissue	23	OG	Microarray + qRT-PCR	[43]
miR-454	Tissue	67	OS	qRT-PCR	[53]
miR-504	Tissue	55	OS	qRT-PCR	[52]
miR-7a, miR-24-3p	Tissue	25/50	PTT	ddPCR	[38,39]
miR-486-5p	Tissue	262	DB	qRT-PCR	[79]
miR-140-5p	Tissue	19	PTT	qRT-PCR	[58]
miR-103	Tissue	32	PTT	qRT-PCR	[56]
miR-219	Tissue	32	OS	qRT-PCR	[51]
miR-451	Tissue	72	OS	qRT-PCR	[50]
miR-375	Tissue	60	DB	qRT-PCR	[80]
miR-675	Tissue	92	OG	qRT-PCR	[42]
miR-34b, miR-34c	Tissue	52	DB	qRT-PCR	[81]
miR-588	Tissue	85	PTT	qRT-PCR	[55]
miR-200a-3p, miR-200a-5p, miR-200b-3p, miR-200b-5p, miR-429	Tissue	1341	DB	qRT-PCR	[82]
miR-663a	Tissue	62	DB	qRT-PCR	[83]
miR-218-5p	Tissue	NS	OS	qRT-PCR	[48]
miR-155	Tissue	1341	DB	qRT-PCR	[84]
miR-203	Tissue	125	DB	qRT-PCR	[85]
miR-34c, miR-183, miR-210	Tissue	103	OS, OG, OG	qRT-PCR	[47]
miR-23b	Tissue	NS	PTT	qRT-PCR	[63]
miR-34b-3p	Tissue	100	PTT	Microarray	[37]

This table summarizes studies focused on the connection between lung cancer and miRNA expression reported after the review by Feng et al. was published [31]. Abbreviations: DB—diagnostic biomarker, OS—onco-suppressor, OG—oncogene, PTT—possible therapeutic target, NS—not specified.

**Table 2 jpm-11-00060-t002:** An overview of studies focused on links between air pollution exposure and miRNA expression.

Pollutant	miRNA	Tissue	Subjects/Characteristics	Method	Reference
**Effects of Environmental Air Pollutants**					
PM2.5, UFP (PM0.1), black carbon, soot	miR-24-3p, miR-4454, miR-4763-3p, miR-425-5p, let-7d-5p, miR-502-5p, and miR-505-3p were associated with PM2.5 exposure	Blood	143, healthy	Microarray	[87]
PM10, PM2.5, PM0.1, HCHO, NO_2_	miR-155 was associated with PM2.5 and HCHO exposure in the asthma group	Serum	180, healthy/180 asthmatic children	qRT-PCR	[88]
PM10, PM2.5, NO, NO_2_, CO, CO_2_, BC and UFP	miR-28-3p, miR-222-3p, miR-146-5p, miR-30b-5p/30c-5p, miR-320a-3p/320b/320c/320d/320e, miR-532-5p, miR-192-5p/215-5p, miR-144-3p, miR-425-5p were associated with exposure to a mixture of pollutants; no effect for PM10 or PM2.5 alone	Plasma	24, healthy	NGS	[89]
PM2.5	Negative link of miR-21, miR-146a and miR-222 expression with PM2.5 in 2nd trimester; positive link of miR-20a and miR-21 in 1st trimester exposure	Placenta	210 newborns	qRT-PCR	[90]
PM2.5	miR-199a/b and miR-223–3p modified links between PM2.5 and systolic blood pressure	Extracellular vesicles	22 healthy elderly	NanoStringnCounter^®^ platform	[91]
PM10, PM2.5, black carbon, ultrafine particles and NO_2_	54 miRNAs associated with exposure	Plasma	24 healthy/ischemic heart disease/COPD	NGS	[92]
PM10, PM2.5, NO, NO_2_, CO, CO_2_, BC and UFP	miR-197-3p, miR-29a-3p, miR-15a-5p, miR-16-5p and miR-92a-3p associated with exposure to pollutants	Blood	50 healthy, 20 COPD, 19 ischemic heart disease	Sureprint G3 Human V19 miRNA 8 x 60K (Agilent)	[93]
PM2.5	The expression of miR-21-5p, miR-187-3p, miR-146a-5p, miR-1-3p, miR-199a-5p was associated with the exposure	Blood	55 healthy	qRT-PCR	[94]
PM10	The expression of let-7c-5p; miR-106a-5p; miR-143-3p; miR-185-5p; miR-218-5p; miR-331-3p; miR-642-5p; miR-652-3p; miR-99b-5p was downregulated	Extracellular vesicles	1630 overweight/obese	QuantStudio™ 12 K Flex Real Time PCR	[95]
PM2.5, UFP	miR-222 expression affected by UFP, but not PM2.5; no effect was observed for miR-146a	Saliva	80 healthy children	qRT-PCR	[96]
PM2.5	Expression of miR-126-3p, miR-19b-3p, miR-93-5p, miR-223-3p, miR-142-3p, miR-23a-3p, miR-150-5p, miR-15a-5p, miR-191-5p, let-7a-5p affected by the exposure	Serum	22 healthy	NanoStringnCounter^®^ platform	[97]
PM and associated metals	17 miRNAs affected by the exposure, including mir-196b, miR-302b, miR-200c, miR-30d	Extracellular vesicles	55 healthy steel plant workers	qRT-PCR	[98]
PM10	miR-145, miR-197, miR-30b, miR-345, miR-26a, miR-425-5p, miR-331, miR-140-3p, miR-101 associated with the exposure	Blood	90 obese subjects	TaqMan Low-Density Array	[99]
PM10	Negative link with miR-21, miR-222, but not -miR-146a expression	Blood	50 healthy adults	qRT-PCR	[100]
PM2.5, metals	Positive link with miR-4516	Serum	120 healthy subjects	miRCURY LNA^TM^	[101]
PM10, PM2.5, elemental carbon	No effect of PM2.5 exposure; PM10 affected 12 miRNAs; EC affected 28 miRNAs in the controls and 29 in truck drivers; miR-125a-5p, miR-1274a, miR-600, miR-1283, miR-10a were common in both groups	Blood	120 healthy subjects (truck drivers and controls)	NanoStringnCounter^®^ platform	[102]
PM	Increased expression of miR-128 and miR-302c	Extracellular vesicles	63 healthy steel plant workers	qRT-PCR	[103]
PM2.5, black carbon, organic carbon, sulfates	Negative links with miR-1, miR-126, miR-135a, miR-146a, miR-155, miR-21, miR-222 and miR-9	Blood	153 elderly healthy men	qRT-PCR	[104]
**Effects of Cigarette/Tobacco Smoke Exposure**					
Biomass smoke (BS), tobacco smoke	miR-22-3p downregulated after BS exposure	Serum	50, COPD	qRT-PCR	[105]
Cigarette smoke	Expression of miR-29a, miR-93, let-7a, and let-7g affected	Serum	775, healthy, smokers/non-smokers	Low-density PCR array	[106]
Cigarette smoke	Modulation of miR-181c expression	Lung tissue	34 COPD	qRT-PCR	[107]
Tobacco smoke	Positive link with miR-223, but not with miR-155	Maternal and cord blood	441 mothers and newborns	qRT-PCR	[108]
Cigarette smoke	28 miRNAs differentially expressed in smokers when compared with non-smokers	Spermatozoa	13 healthy smokers and non-smokers	miRCURY LNA^TM^	[109]
**Effects of Other Pollutants**					
Wood and LPG exhaust	miR-126 and miR-155 upregulated after wood smoke exposure	Plasma	52, healthy	qRT-PCR	[110]
Diesel exhaust, allergen	miR-183-5p, miR-324-5p and miR-132-3p induced by allergen; no modulatory effect of diesel exhaust	Bronchoalveolar lavage	15 atopic subjects	NanoStringnCounter^®^ platform	[111]
VOC	Specific miRNAs for exposure to individual VOCs	Blood	50 healthy exposed workers	Microarray	[112]
Ozone	Increased expression of miR-132, miR-143, miR-145, miR-199a, miR-199b-5p, miR-222, miR-223, miR-25, miR-424, and miR-582-5p after the exposure	Sputum	20 healthy volunteers	Microarray	[113]

**Table 3 jpm-11-00060-t003:** A summary of miRNAs identified to be deregulated in lung cancer, as well as in air pollution-exposed subjects.

miRNA	Pollutant	References-Air Pollutants	References-Lung Cancer
**miR-222**	**mixture of pollutants; PM2.5; UFP; PM10; black carbon, organic carbon, sulphates; ozone**	[89,90,96,100,104,113]	[114]
**let-7a-5p**	**PM2.5; Cigarette smoke**	[87,97]	[64]
**miR-21**	**PM2.5; PM10; black carbon, organic carbon, sulphates**	[90,94,100,104]	[41,68,74,77]
**miR-29a family**	**Cigarette smoke**	[93,106]	[44]
**miR-93 family**	**PM2.5; Cigarette smoke**	[97,106]	[36]
**miR-126 family**	**PM2.5; black carbon, organic carbon, sulphates; wood smoke**	[97,104,110]	[65,70]
**miR-145**	**PM10; ozone**	[99,113]	[65]
**miR-155**	**PM2.5, HCHO; black carbon, organic carbon, sulphates; wood smoke**	[88,104,108,110]	[35,84]
**miR-223**	**PM2.5; tobacco smoke; ozone**	[91,97,108,113]	[40]
**miR-425 family**	**PM2.5; mixture of pollutants; PM10**	[87,89,99]	[36,69]
miR-1-3p	PM2.5	[94]	[64]
miR-19bfamily	PM2.5	[97]	[36]
miR-22-3p	Biomass smoke (BS), tobacco smoke	[105]	[36]
miR-24-3p	PM2.5, UFP (PM0.1), black carbon, soot	[87]	[38,39]
miR-25	Ozone	[113]	[66,75]
miR-101	PM10	[99]	[61]
miR-142 family	PM2.5	[97]	[36]
miR-183 family	Diesel exhaust, allergen	[111]	[47]
miR-185	PM10	[95]	[36]
miR-196b family	PM and associated metals	[98]	[62]
miR-200c	PM and associated metals	[98]	[35,67]
miR-218-5p	PM10	[95]	[48]
miR-532	PM10, PM2.5, NO, NO_2_, CO, CO_2_, BC and UFP	[89]	[69]
miR-642 family	PM10		[35]
miR-1274a	PM10, PM2.5, elemental carbon	[102]	[36]

Text in bold red: miRNA commonly deregulated in six air pollution studies; text in bold: miRNAs commonly deregulated in 2–4 air pollution studies; regular font—miRNAs unique for a single air pollution study. See Figure 1 for graphical presentation. This overview reflects lung cancer studies reported since the Feng et al. review [31].

## Data Availability

Not applicable.

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
