# Peer review of "The Impact of Air Pollution Exposure on the MicroRNA Machinery and Lung Cancer Development"

_jpm, 2021, doi:10.3390/jpm11010060_

Round 1

Reviewer 1 Report

Sima et al submit a review article entitled "The impact of air pollution exposure on the microRNA machinery and lung cancer development"

They searcheds the papers studying miRNA levels affected by environmental pollutants, including airborne contaminants. They conclude that a total of 25 miR-NAs are dysregulated, among them, miR-222, miR-21, miR-126-3p, miR-155 and miR-425 being the most prominent. These miRNAs could be biomarkers candidates for the identification of human populations exposed to air pollution with a significant risk of developing lung cancer.

Introduction:

a short paragraph on the meaning an differences between -5p and -3p would be useful

When talking about the RT-qPCR method the authors should seaprate cybergreen and taqman and explain why taqman is more precise. Also it is mandatory to discuss the role and importance of reporter gene.

The authors should elaborate what they mean by "The hybridization method". If this is microarray, it is mostly outdated by now.

I have noticed that miR-223 is also mentionned in a lot of studies, so perhaps it could be added to the short list?

miR-126 and miR-222 are involved in angiogenesis in a lot of studies, perphaps this should be discussed in the contect lof lung cancer? See for example DOI: 10.2174/1570161114666160914175149

a figure wiht a picure of lung vessels etc and the most relevant miRNAs would be useful

Minor

In introduction correct "500-3,000 base long molecule " to "500-3,000 bases long molecule ".

What exactly does " In order not to repeat the previously summarized data, from the aforesaid review [" mean?

In "3. Air pollution and miRNA In comparison with (lung) cancer", why do the authors use brackets

page 13 change mir-196b to miR-196b (unless it tis the pre-MIR that is affected?)

Author Response

We thank the reviewers for their valuable comments that helped to improve our manuscript. Our response is in the text below, the modified text is highlighted in the manuscript.

Reviewer 1:

Sima et al submit a review article entitled "The impact of air pollution exposure on the microRNA machinery and lung cancer development"

They searched the papers studying miRNA levels affected by environmental pollutants, including airborne contaminants. They conclude that a total of 25 miR-NAs are dysregulated, among them, miR-222, miR-21, miR-126-3p, miR-155 and miR-425 being the most prominent. These miRNAs could be biomarkers candidates for the identification of human populations exposed to air pollution with a significant risk of developing lung cancer.

Introduction:

1. A short paragraph on the meaning an differences between -5p and -3p would be useful

Response: We prepared a short paragraph (the last paragraph in Introduction, section 1.1) that summarizes miRNA nomenclature, including explanation of differences between -5p and -3p miRNA variants.

2. When talking about the RT-qPCR method the authors should separate cybergreen and taqman and explain why taqman is more precise. Also it is mandatory to discuss the role and importance of reporter gene.

Response: The text describing the differences between both variants of qRT-PCR is added to Introduction (section 1.2). We also mention the role of a reporter (reference) gene.

3. The authors should elaborate what they mean by "The hybridization method". If this is microarray, it is mostly outdated by now.

Response: Yes, this method is based on microarrays. We added the explanation to the text (Introduction, section 1.2). We agree that microarrays are being replaced by other methods (e.g. NGS). The limitation of the method is mentioned in the text.

4. I have noticed that miR-223 is also mentioned in a lot of studies, so perhaps it could be added to the short list?

Response: We agree that this molecule appears in many air pollution studies. However, based on our search criteria, we identified only one study linking miR-223 with lung cancer. Thus, we did not show it in the shortlist. Nevertheless, we still show miR-223 in Table 3 and Figure 1 as one of the prominent deregulated molecules.

5. miR-126 and miR-222 are involved in angiogenesis in a lot of studies, perphaps this should be discussed in the context of lung cancer? See for example DOI: 10.2174/1570161114666160914175149

Response: We thank the reviewer for this comment. The information on angiogenesis, along with the reference, was added to section 2.

6. a figure with a picture of lung vessels etc and the most relevant miRNAs would be useful

Response: We agree that a picture is useful to completement information in the text. We checked the current Figure 1 and found that its content partly overlaps with information suggested by the reviewer. We thus modified the figure to better graphically represent the target organ (lungs) and sources of air pollution.

Minor

In introduction correct "500-3,000 base long molecule " to "500-3,000 bases long molecule ".

Response: The typo was corrected.

What exactly does " In order not to repeat the previously summarized data, from the aforesaid review [" mean?

Response: In our review we tried to avoid repeating information published by Wozniak et al., PLoS One, 2015. Thus, our manuscript deals with studies not covered by that review. The text was slightly modified to make it clearer.

In "3. Air pollution and miRNA In comparison with (lung) cancer", why do the authors use brackets

Response: The meaning was “lung or other cancer”. We removed brackets to avoid confusion.

page 13 change mir-196b to miR-196b (unless it tis the pre-MIR that is affected?)

Response: We checked the original article – mir-196b is correct.

Reviewer 2 Report

The manuscript "The impact of air pollution exposure on the microRNA machinery and lung cancer development" presented by Michal Sima et al., was aimed to review the current research and clinical studies for miRNA expression profiles associated with lung cancers and  miRNAs deregulated after exposure to air pollutants, and to identify miRNAs commonly deregulated under both conditions.

Based on the analyzed data of published works, the authors found 25 miR-NAs meeting these criteria and proposed these miRNAs as biomarkers of choice for the identification of human populations exposed to air pollution with a significant risk of developing lung cancer.

The review provides detailed analysis of outcomes for numerous studies of miRNA expression profiles associated with lung cancers and miRNAs deregulated after exposure to different types of air pollutants, which make main contribution to lung disease development.

In my opinion, this review article may be interesting and helpful for a wide circle of researchers and physicians, as well as university students. The information provided in this manuscript may be useful for further environmental and toxicological research. This manuscript is well organized and comprehensively described. The references were used properly. I have following minor comments and suggesting to the manuscript:

  1. To improve readability and comprehension of the text, it is necessary to divide the long Chapter 3 into subsections. For example, miRNA expression affected by air pollution exposure in adults, children, in elderly humans, in tobacco smoking subjects, in occupationally exposed subjects, etc.
  2. Throughout the text, it is better to use 'positive correlation', 'positive link' and 'positive relationship' instead of 'positive association'.

Author Response

We thank the reviewers for their valuable comments that helped to improve our manuscript. Our response is in the text below, the modified text is highlighted in the manuscript.

1. To improve readability and comprehension of the text, it is necessary to divide the long Chapter 3 into subsections. For example, miRNA expression affected by air pollution exposure in adults, children, in elderly humans, in tobacco smoking subjects, in occupationally exposed subjects, etc.

Response: We agree with the reviewer that the text is difficult to follow. We divided the chapter as suggested.

2. Throughout the text, it is better to use 'positive correlation', 'positive link' and 'positive relationship' instead of 'positive association'.

Response: We replaced “association” with the suggested terms throughout the text.

Round 2

Reviewer 1 Report

changes are ok